# Genome-Wide Identification and Analysis of the Aureochrome Gene Family in *Saccharina japonica* and a Comparative Analysis with Six Other Algae

**DOI:** 10.3390/plants11162088

**Published:** 2022-08-11

**Authors:** Yukun Wu, Pengyan Zhang, Zhourui Liang, Yanmin Yuan, Maohong Duan, Yi Liu, Di Zhang, Fuli Liu

**Affiliations:** 1College of Fisheries and Life Science, Shanghai Ocean University, Shanghai 201306, China; 2Key Laboratory of Sustainable Development of Marine Fisheries, Ministry of Agriculture and Rural Affairs, Yellow Sea Fisheries Research Institute, Chinese Academy of Fishery Sciences, Qingdao 266071, China; 3Key Laboratory of Marine Genetics and Breeding, Ministry of Education, College of Marine Life Science, Ocean University of China, Qingdao 266100, China

**Keywords:** aureochrome (AUREO), blue photoreceptor, bZIP, genome-wide analysis, AUREO gene family, *Saccharina japonica*

## Abstract

Aureochrome (AUREO) is a kind of blue light photoreceptor with both LOV and bZIP structural domains, identified only in Stramenopiles. It functions as a transcription factor that responds to blue light, playing diverse roles in the growth, development, and reproduction of Stramenopiles. Most of its functions are currently unknown, especially in the economically important alga *S. japonica* farmed on a large scale. This study provided a comprehensive analysis of the characteristics of AUREO gene families in seven algae, focusing on the AUREOs of *S. japonica*. AUREO genes were strictly identified from seven algal genomes. Then AUREO phylogenetic tree was constructed from 44 conserved AUREO genes collected. These AUREO genes were divided into five groups based on phylogenetic relationships. A total of 28 genes unnamed previously were named according to the phylogenetic tree. A large number of different cis-acting elements, especially bZIP transcription factors, were discovered upstream of AUREO genes in brown algae. Different intron/exon structural patterns were identified among all AUREOs. Transcriptomic data indicated that the expression of *Sj* AUREO varied significantly during the different development stages of *S. japonica* gametophytes. Periodic rhythms of light induction experiments indicate that *Sj* AUREO existed in a light-dependent circadian expression pattern, differing from other similar studies in the past. This may indicate that blue light affects gametophyte development through AUREO as a light signal receptor. This study systematically identified and analyzed the AUREO gene family in seven representative brown algae, which lay a good foundation for further study and understanding of AUERO functions in agal growth and development.

## 1. Introduction

Light is both an energy source and an important signal source for organisms on earth. To efficiently utilize the light information, eukaryotic cells have evolved a complex photosensitive system in which photoreceptors selectively absorb various parts of the solar spectrum and act as the first step in light signal transduction [1,2]. Photoreceptors endow photosynthetic autotrophs with the ability to continuously respond to light and allow them to adapt their reproduction, growth, development, and behavior accordingly [3].

The shallow waters of the intertidal region are an attractive habitat for marine sedentary photosynthetic organisms, providing them with both a substratum and access to light. However, the nearshore water environment is also a hostile environment which necessitates an ability to cope with the biotic stresses of the organisms [4]. As the short wavelength of UV-B and long wavelength of red light are mostly absorbed by water, blue light dominates the marine water environment. Therefore, blue light is an important environmental factor for benthic organisms. For example, it is essential to induce kelp gametophyte development [5].

The photosynthetic heterokonts share an ancestral endosymbiotic event of phagocytosis of a red alga, giving birth to their plastids [6,7]. While the relationships among many classes of Ochrophyta remain unresolved, three main groups (SI, SII, SIII) are supported in most phylogenies. The brown algae are situated within lineage SI as part of a radiation of classes during the late Paleozoic [7]. Kelps are ecologically important primary producers and ecosystem engineers, playing roles in structuring nearshore habitats; for example, nutrient cycling, energy capture and transfer, and providing biogenic coastal defense [8]. Kelps dominate rocky reefs in lower intertidal and shallow subtidal zones throughout temperate and subpolar regions of the world. Given that kelp forests dominate approximately 25% of the world’s coastlines, the global value of ecosystem services provided by brown algae is liable to be in the hundreds of billions of USD per year [7]. Species of Laminaria and Saccharina are the most noteworthy macroalgal genera in the kelp, their life history one of alternating between large sporophyte (diploid) and microscopic gametophyte generations [9]. They are naturally distributed in coastal ecosystems in temperate and subpolar regions of the globe, especially in the northern hemisphere [5]. Research in algal physiology over the last century has indicated that the effect of light on the growth and development of kelp is pivotal, especially blue light [10] and photoperiod [11].

To date, 13 photoreceptors have been identified in Arabidopsis. Among them, cryptochromes (cry1, cry2) monitor the blue/ultraviolet A region of the spectrum, phytochromes (phyA–phyE) monitor the red/far-red region, and UVR8 monitors the ultraviolet B region. The blue light (BL) photoreceptor-cryptochrome (CRY) was first identified in Arabidopsis [12], and later a blue light receptor with phototropic properties was identified and named phototropin [13,14]. Current research suggests that CRY plays a crucial role in the growth and development of terrestrial plants [15], including entrainment of the circadian clock [16], coordination of temperature sensing [17], and the regulation of photoresponsive transcription [15]. The blue light photoreceptor CRY is considered the earliest photoreceptor of origin [18]. However, only a handful of marine algal cryptochromes have been characterized spectroscopically and photochemically [19]. Furthermore, the marine algal characterized cryptochromes lack the C-terminal domain typical of plant CRY that enables interaction with partner molecules, suggesting that different signaling mechanisms could be used in marine algae [20]. While various BL receptors have been discovered in green plants and other organisms, BL receptors in photosynthetic stramenopiles remained unknown until 2007 [21].

Aureochromes (AUREOs) were identified firstly in two algae of stramenopiles, i.e., *Vaucheria frigida* and *Fucus distichus* [21]. AUREOs only have been detected in stramenopiles to date [22,23]. Brown algae, as one taxon of stramenopiles, originated as a secondary endosymbiosis of unicellular red algae in eukaryotic host cells [24], which are very distantly related to the most intensively studied eukaryotic species to date [25]. This independent evolutionary journey has endowed brown algae with many new metabolic, physiological, cellular, and ecological characteristics [17,26]. Developmental processes are particularly interesting in this group, as they have evolved complex multicellular structures independently of animals, fungi, and green plants, but this also represents a barrier, as there is only limited correlation between brown algae and model species such as Arabidopsis [26].

AUREOs possess both a basic leucine zip (bZIP) structural domain for DNA binding and a light-oxygen-voltage (LOV) structural domain for light reception. In *V. frigida*, *Vf* AUREO1 and *Vf* AUREO2 are located in the nucleus as homo- or heterodimers, and both are involved in the photomorphogenic response of *V. frigida* [21]. A total of four AUREOs were discovered in *Phaeodactylum tricornutum* [22], exhibiting a rhythmic oscillatory expression pattern during the 24 h cycle. *Pt* AUREO1a and *Pt* AUREO1c have a light-independent circadian regulation pattern. *Pt* AUREO1a can be bound to G-box with ACGT core sequence [27]. Light causes a conformational change in the *Pt* AUREO1a homodimeric protein complex to increase the affinity of the bZIP domain to the dsCYC2 promoter, allowing it to bind to different regulatory elements in the Diatom-Specific Cyclin (dsCYC2) promoter. This ultimately activates dsCYC2 and initiates the cell cycle [28]. While 75% of genes in *P. tricornutum* were abundantly expressed within 1 h after transfer from red to blue light, these genes were completely suppressed in independent knockout lines of *Pt* AUREO1a, indicating that *Pt* AUREO1a is a highly efficient blue light switch [29].

Recent studies showed that the AUREOs probably play important roles in regulating blue light effects on *S. japonica*. Five *Sj* AUREOs were discovered in the *S. japonica* transcriptome shortly after the AUREO discovery in *V. frigida* [30]. The 40S ribosomal protein S6 (RPS6) and miR8181 were involved in the regulation of AUREO in *S. japonica*, playing roles in cellular division and photomorphogenesis [31] and negatively regulating *S. japonica* growth and development [32]. However, AUREOs represent a new type of photoreceptor, and most of its functions are currently unknown [33], especially in algae. For a long time, we have lacked a comprehensive and systematic understanding of the *Sj* AUREO genes [30,31,32]; for example, the number and chromosome location of the AUREO genes in *S. japonica* remain unclear.

Fortunately, the genome data available of several brown algae provide new opportunities to identify the AUREOs in these algae [4,34,35,36,37,38,39]. In this study, based on the genome sequence data, the AUREO gene family in seven algae was identified and characterized, focusing on the AUREO gene family form *S. japonica*. The uncovered information about AUREOs in this study will open a new gateway for subsequent AUREO studies in brown algae.

## 2. Results

### 2.1. Identification and Characterizing of Aureochrome in Seven Algae

A total of 44 AUREO genes were identified and collected based on the seven published algal genomes (*V. frigida*, *E. siliculosus*, *S. japonica*, *U. pinnatifida*, *C. okamuranus*, *N. decipiens*, and *P. tricornutum*) to date (Appendix A). The AUREO genes were divided into five major types and named according to the available nomenclature information and phylogenetic relationship. The key information of these AUREO genes is listed in Appendix A. Compared to the reported two and four AUREO genes identified in two other algae, V. frigida and *P. tricornutum* [21,22], five, nine, eight, seven, and nine AUREO genes were identified respectively in the brown algae of *E. siliculosus*, *S. japonica*, *U. pinnatifida*, *C. okamuranus* and *N. decipiens* in this study. The AUREO genes were not clustered but scattered on different chromosomes or scaffolds in brown algae. The ORF length of AUREO genes varied significantly among genes within or among species, with the longest one *Nd* AUREO2b of 4323 bp and the shortest one *Ec* AUREO3 of 873 bp. The gene structures also varied significantly among genes within or among species; for example, *Up* AUREO1a has 10 exons, while *Pt* AUREO1a has just two exons. The PI of AUREO proteins ranged from 4.66 to 10.52, while MV ranged from 24.3 kDa to 155.1 kDa, indicating a marked difference in the basic properties of the AUREO proteins within or among species.

The results for the transmembrane helix (TMH), number of phosphorylation sites, and subcellular localization of AUREO were listed in Appendix A. Built on the predictions of TMHMM-2.0, almost all the identified AUREOs have no transmembrane structures, while *Nd* AUREO2b had one and *Co* AUREO5 has two transmembrane structures. The phosphorylation of AUREOs prefers to occur on the Serine sites than on other amino acids (aa). *Sj* AUREO1b and *Nd* AUREO1c have the greatest phosphorylation sites of 114. The subcellular localization analysis showed that most of the 44 AUREO proteins were predicted to be localized in the nucleus, with only five localized in the chloroplast. The two main protein folding modes, alpha helix and beta turn, accounted for 1.75–55.61% and 29.60–55.63%, respectively. They form the principal folding modes in the AUREO protein with a proportion of 68.72–84.71%. The other two folding methods, beta turn and extended strand, accounted for 3.42–13.20% and 8.18–19.16%, respectively (Appendix A).

### 2.2. Characteristics of Structural Domain and Gene Structure Evolution of Aureochrome

Figure 1 illustrates the specific location of the bZIP and LOV structural domains in *Sj* AUREOs. The bZIP and LOV structural domains of most *Sj* AUREOs are generally located nearer to the N-terminal (except the *Sj* AUREO1c, it’s almost completely close to the C-terminal), and the LOV domain was much closer to the C-terminal. In *Sj* AUREO1b/1c/1e the two structural domains were closely connected, while in the other five AUREOs they were separated by 45–97 amino acids.

The bZIP and LOV structural domains of several algal species shared a high degree of similarity. According to the pfam-derived data and sequence comparison, the homologous parts of the bZIP and LOV structural domains in different AUREOs were 45 aa and 111 aa (Figure 1), respectively. Figure 1B illustrates the results of sequence alignment of *Sj* AUREO with a fragment of bZIP in Arabidopsis, where only the DNA-bound basic region (1–7 aa) of the bZIP structural domain was intercepted with the leucine zipper structure. The basic regions of *Sj* AUREO1a/3/4/5 (Figure 1B, 1–7 aa) have high similarity to groups G, C, and S in AtbZIP. In the B/C/G/S group of Arabidopsis bZIP (AtbZIP), the third and fourth amino acids in the N-GB 7-R motif are negative charge Glu3 and polar Ser4. In contrast, in the bZIP domain of *Sj* AUREO, these two amino acids were converted to negative charge Glu3 and aromatic His4 (*Sj* AUREO1a/3/4/5), hydrophobic Val3 and Leu4 (*Sj* AUREO2b), negative charge Asp3 and aromatic His4 (*Sj* AURE1d/1e), positive charge Lys3 and polar Ser4 (*Sj* AUREO1b/*Sj* AUREO1c). The latter two groups are varied by not being found in several diatoms and *E. siliculosus*. Repeated hydrophobic amino acid structures at positions 19/26/33 in *Sj* AUREO bZIP can be clearly observed (Figure 1B), and they are typical of the structures that constitute the leucine zipper.

To gain insight into the gene structure of the AUREO genes from Phaeophyta, their exon/intron organization was analyzed (Figure 2). The number of exons within the AUREO gene tegument of the three brown algae ranged from three to ten, with an average of seven. The largest number of exon structures were *Sj* AUREO1a and *Up* AUREO1a with 10 exons, while the smallest was *Up* AUREO2 with only 3 exon junctions. Despite interspecific differences, the structure of the intron exons of the three brown algals (*E. siliculosus*, *S. japonica*, *U. pinnatifida*) AUREO genes differed somewhat more markedly between subgroups than between species within the same subgroup of AUREO.

### 2.3. Phylogenetic Analysis and Classification of the AUREO Gene Family

To further explore the evolutionary relationship of AUREO genes between *S. japonica* and other six stramenopiles species, an unrooted phylogenetic tree was constructed based on the 44 AUREO proteins from the above seven algae. The 16 previously named AUREO proteins were marked in red. The newly identified 28 AUREO genes (four of *S. japonica*, eight of *U. pinnatifida*, seven of *C. okamuranus*, and nine of *N. decipiens*) were named according to their evolutionary relationship revealed by the phylogenetic tree (Figure 3), with a revision of *Sj* AUREO1 into *Sj* AUREO1a [33]. Based on 16 AUREO proteins identified previously, the identified ARUEO were clustered into five groups (Figure 3). Group I was further divided into three subgroups, i.e., subgroup IA, subgroup IB, and subgroup IC. Subgroup IA (yellowish) was the largest one, including two AUREOs from *C. okamuranus*, four from *S. japonica*, three from *U. pinnatifida*, two from *N. decipiens,* and one from *P. tricornutum*. Subgroup IB and subgroup IC contained four and five AUREOs, respectively. For the other four AUREO groups, group II contained 8 AUREOs of the AUREO2 subfamily, group III contained 6 AUREOs of the AUREO3 subfamily, group IV contained 4 AUREOs of the AUREO4 subfamily, and group V contained 5 AUREOs of AUREO5 subfamily. These four groups were clustered in another large subfamily distinct from the AUREO1 subfamily, with all three large brown algae (*E. siliculosus*, *S. japonica*, *U. pinnatifida*) present with closely related AUREO2/3/4/5 protein members with high confidence (Bootstrap value >98). A total of nine AUREO proteins were identified in *S. japonica*, and they were distributed in all subgroups except subgroup IB, among which subgroup IA was the most distributed, with a total of 4 AUREOs proteins.

### 2.4. Analysis of Cis-Acting Elements of AUREO Gene Upstream

The cis-acting elements were identified from the 2000 bp upstream of *Sj* AUREOs (Figure 4). From the 34 classes of elements, six of the more critical promoters were selected and plotted (Figure 4). Light-responsive elements were identified in all the *Sj* AUREOs. *Sj* AUREO2 was special in that only meristem expression and light-responsive elements among the six types of elements were present in it. Low-temperature responsiveness and potential bZIP recognition sites containing ATCG sequences [41,42] were present upstream of most *Sj* AUREOs. All the original labels of potential recognition sites of bZIP were light response elements The transient action element site of cell cycle regulation site only occurs in *Sj* AUREO1e, and circadian control only occurs in *Sj* AUREO1a/5. However, the meristem expression-related cis-acting element was not found in *Sj* AUREO5.

### 2.5. Expression Analysis of AUREO Genes at Different Developmental Stages of S. japonica Gametophyte

To explore the effect of *Sj* AUREOs on gametophyte growth and development, the expression levels of *Sj* AUREOs at different growth and developmental stages of gametophytes were quantified based on transcriptome data (Figure 5). Overall, the expression of most *Sj* AUREOs (except *Sj* AUREO1e and *Sj* AUREO5) showed a downward trend when the gametophyte transitioned from a dormant state (on day 0) through vegetative growth (on day 3) to a reproductive development state (on day 6 and day 9). Among them, *Sj* AUREO2 had the most significant downward trend, and its expression level in 0 d was more than 40 times that on day 9. The expression levels of *Sj* AUREO1e and *Sj* AUREO5 showed a trend of increasing first and then decreasing. The highest expression level of *Sj* AUREO1e (on day 6) was 22 times higher than that of the lowest expression level (on day 9). The expression levels of different *Sj* AUREOs in gametophytes varied greatly, among which *Sj* AUREO1c (highest FPKM values >400; FPKM: fragments per kilobase per million mapped fragments) and *Sj* AUREO5 (the highest FPKM value is close to 100) had the highest transcript abundance, while *Sj* AUREO1e (the highest FPKM value is less than 4) and *Sj* AUREO3 (the highest FPKM value is about 12) have lower transcript abundance. The variation in the expression level of *Sj* AUREOs indicated that different *Sj* AUREOs might play different functions in gametophyte growth and development.

### 2.6. Circadian Rhythm of AUREO Expression in S. japonica

To investigate the spatio-temporal pattern of AUREO expression of *S. japonica* gametophyte, light- and circadian-dependent transcript expression was analyzed using qPCR (Figure 6). The results showed that all *Sj* AUREO showed a certain circadian rhythm and a similar expression pattern. Most *Sj* AUREO (except for *Sj* AUREO1c, which peaks at 16 o’clock the next day) reached their peak expression at 12 o’clock on the first day of the experiment. The expression of most *Sj* AUREO decreased to very low levels before entering darkness, except for *Sj* AUREO1c, which fell to this level only after four hours. It was noteworthy that *Sj* AUREO1e has a similar expression peak to *Sj* AUREO1c, although it also peaked at midday on day 4 but expressed at extremely low levels on day 5. The expression peak of most *Sj* AUREOs on the second day was much lower than that on the first day, and the peak was delayed until 4 pm. *Sj* ARUEO1c was a special case in that it peaked at 16:00 and was the highest among all *Sj* AUREOs, although it tended to decline before darkness. Unlike the other *Sj* AUREOs, *Sj* AUREO1c expression did not drop to the same low level as during the dark cycle, and the end of the decline was carried forward until 0:00.

## 3. Discussion

### 3.1. Characteristics of AUREO Domain and Gene Structure Evolution in Algae

Phylogenetic studies could trace the bZIP’s domain back to immediate ancestors in green algae [43]. A classification of bZIP based on homology of the basic region and additional conserved motifs has been proposed in *Arabidopsis* [40]. Following the previous nomenclature [44], bZIPs were classified into 13 groups (designated A-M). Structural data of bZIP demonstrate that only five aa of each base domain facilitates the contact with DNA, and an invariant N-X7-R/K motif with asparagine (N) and basic (R/K) residues with exact spacing was found in the study of model plants [43]. The bZIP in *Sj* AUREO was essentially consistent with this formula. However, there were a few exceptions to this in both *Arabidopsis* and *S. japonica*, and the possible functional relevance of these alterations has not been reported. Similar to previous reports on AUREO [22], there are just three sets of repeated leucine residues in the bZIP domain of AUREO in *S. japonica*. However, this set of repeats is between three and eight in *Arabidopsis* [43].

Homology of the elementary region of bZIP in algae and *Arabidopsis* implies that they share the same DNA recognition site. However, there are also some differences between bZIP of algae and *Arabidopsis*, indicating AUREO performs variable functions and roles in the algae and green plant *Arabidopsis*. It is worth noting that bZIP groups B and C are considered essential in AtbZIP studies due to their conserved nature. These bZIP genes have been identified in most eukaryotes, and their ancestry can probably be traced back to the endosymbiotic event in red algae [42]. The bZIP structural domain in *Sj* AUREO1b\c has high homology with groups B and C of bZIP. In all *Sj* AUREO, *Sj* AUREO1c is thus probably the most critical. The extremely unique specific expression variation of *Sj* AUREO1c appears to corroborate this view.

### 3.2. Evolution and Number of AUREOs from Laminariales and Other Stramenopiles

Positive selection is responsible for the retention of duplicated genes and the consequent expansion of gene families [43,45]. Habitat differences are probably a significant primary driver of the quantitative and evolutionary divergence of AUREO in stramenopiles. The freshwater58 environment is generally stabilized compared to the marine environment [46]. Thus, just two AUREOs are probably sufficient to deal with the light changes for *V. frigida* in freshwater environments. Diatoms handle light signals flexibly compared to some brown algae with alternation of generations. Observations in the natural environment indicate that *P. tricornutum* is a coastal marine diatom that can adapt to unstable environments [47]. *P. tricornutum* can be maintained in the optimum light environment by vertical movement. This may explain why *P. tricornutum* evolved more AUREOs than *V. frigida*.

Light of different wavelengths in the spectrum penetrates seawater at different depths [48]. Red light (~650 nm) is easily absorbed in the ocean surface, while blue light (~450 nm) is extremely penetrating and usually decays to 10% at depths of 82 m below the ocean surface [49]. Thus, the light environment in the coastline ecosystem is very complex. This sophisticated coastline environment exposes native organisms to greater stress [4]. Marine algae species may evolve more photoreceptors to form composite light signal transduction pathways to cope with more complex environments.

Ectocarpales and Laminariales belong to the two newly evolved orders in Phaeophyceae, and they are relatively closely related [4,35]. However, there are certain differences between the AUREO genes of *E. siliculosus* and *S. japonica* in the evolution and quantity. In particular, *E. siliculosus* did not evolve into the large AUREO1 family. *S. japonica* lives in a deeper environment than *E. siliculosus*, which grows on low tidal rocks, in rock pools, or on other seaweeds [50], making it more difficult to capture and identify light sources in different environments. The complex habitat forces *S. japonica* to evolve more photoreceptors and to be more precise in its grasp and recognition of dynamic changes in the light environment as a basis for regulating development and growth strategies.

### 3.3. Rhythmic Expression Pattern of AUREOs and the Possible Function in S. japonica

The crucial role of blue light in the development of kelp gametophytes has been demonstrated as far back as the last century [51]. In a century of kelp light-related research, most of the evidence proved that periodic rhythmic blue light is effective in promoting gametophyte development, while red light almost completely inhibits this process and instead promotes cell division in kelp gametophytes [52,53]. There are two central concerns: (1) gametophyte cell development usually occurs within 30 min after switching from photoperiod to dark cycle and continuous, and (2) uninterrupted exposure to either type of light inhibits the developmental process [10]. These physiological studies indicate that light quality and photoperiodic rhythms are essential for gametophyte development in *S. japonica*.

In stramenopiles, AUREO serves not only as a photoreceptor but also as a transcription factor, as it has both a LOV domain for receiving light signals and a bZIP domain for regulating gene expression. This is distinct from the set of mutually independent regulatory pathways of the light signal in *Arabidopsis*. For example, the CRY -COP1-SPA (Cryptochrome- Constitutive Photomorphogenic 1- Suppressors of Phytochrome A) interaction positively regulates the abundance of the HY5 (Elongated Hypocotyl 5, A member of the bZIP transcription factor family protein) during blue light signal sensing and transduction [15].

The expression pattern of AUREO genes revealed in this study also corroborated the findings of the previous physiological studies. All *Sj* AUREO appeared to have a light-dependent expression pattern and exhibited a regular expression pattern closely related to the biological clock. In earlier studies, two aureochromes regulated by a circadian rhythm and capable of forming heterodimer complexes were found in *P. tricornutum* [27]. Nevertheless, two *Pt* AUREOs were a light-independent pattern of expression [27]. Interspecific and morphological differences may be an important reason for this discrepancy, but experimental differences cannot be ignored. The main difference between the two studies was in the photoperiod, with *Pt* AUREO utilizing a long photoperiod (16 h light) and *Sj* AUREO using a shorter daily light period. AUREO may adopt different patterns to adapt to different living environments, in line with the annual variation in *S. japonica’s* adaptation to the intertidal zone [53].

There was a certain difference in the expression of AUREO genes in the process of gametophyte growth and development, especially when it develops on the ninth day (some gametocytes have released eggs at this time point). Combined with the phenomenon observed in physiological experiments and the critical function of photoreceptors in plant morphological changes in higher plants, we infer that there may be a close relationship between AUREO and gametophyte development in kelp. Nonetheless, the signaling pathways regulating blue light in *S. japonica* remain largely elusive to date. *Sj* AUREO probably plays a more significant role in developing the gametophyte than simply acting as a receptor for receiving light signals because of the bZIP structural domain in *Sj* AUREO. It has been noted that bZIP TFs are involved in developmental and physiological processes as well as biotic/abiotic stress responses under normal and stressed growth conditions. They are essential for various plants to withstand adverse environmental conditions [54,55,56]. This implies that AUREO genes with a bZIP structural domain may have a similar role. Thus, the function of the bZIP domain of AUREO may be an important direction for future study on the algal photoreceptor AUREOs.

## 4. Materials and Methods

### 4.1. Data Collection

Download currently published algal genome data from public databases: *E. siliculosus* (https://bioinformatics.psb.ugent.be/gdb/ectocarpusV2/, accessed on 21 July 2021) [4], *S. japonica* (https://bioinformatics.psb.ugent.be/gdb/Saccharina/, accessed on 21 July 2021) [35], *U. pinnatifida* (https://figshare.com/s/94aebbd77f374b9c6faf, accessed on 28 July 2021) [38,39] (http://www.magic.re.kr/portal/assembly/MA00358), *P. tricornutum* [34] (https://phycocosm.jgi.doe.gov/pages/search-for-genes.jsf?organism=Phatr2_bd, accessed on 21 July 2021). *C. okamuranus* [36] (https://academic.oup.com/dnaresearch/article/23/6/561/2647449?login=true, accessed on 10 September 2021), *N.*
*decipiens* [37] (https://www.nature.com/articles/s41598-019-40955, accessed on 10 September 2021).

The AUREOs identified in *P. tricornutum* and *V. frigida* [21] were extracted from earlier published papers, while other AUREOs are first obtained by importing their pfam numbers into the genome and pfam database (http://pfam.xfam.org/, accessed on 11 Novembe 2021). BZIP and PAS (Per-ARNT-Sim. The LOV family was merged into a larger PAS family in the pfam database) conserved domain mosaic on AUREO. Download the latest Pfam full-mode database (http://ftp.ebi.ac.uk/pub/databases/Pfam/current_release/Pfam-A.hmm.gz, accessed on 11 Novembe 2021). Finally, the pfam database, the genomic data, and the pfam numbers of the two conserved domains were imported into the Simple HMM Wrapper in Tbtools [57] (https://github.com/CJ-Chen/TBtools/releases; Version number: 1.9876.0.0; Creator: Chen Chengjie; Guangzhou, China) software. Conserved domain prediction of the AUREO gene in the found brown algae was performed by rapid identification of conserved structural domains in protein sequences by RPS-BLAST using CDD in NCBI (https://www.ncbi.nlm.nih.gov/cdd/?term=, accessed on 11 Novembe 2021) and exogenous databases to delete some uncertain sequences (Appendix A). ClustalW (https://www.genome.jp/tools-bin/clustalw; Creator: Julie D. Thompson Tokyo, Japan, accessed on 21 Novembe 2021) was used to perform multiple sequence comparisons of the intercepted bZIP and PAS fragments (Appendix A), and presented the comparison results via Jalview (http://www.jalview.org/getdown/release/; Version number: 2.11.2.0; Creator: Andrew M. Waterhouse; Dundee; British).

### 4.2. Structural and Phylogenetic Analysis of AUREO

PI/MW prediction for AUREO using Expasy (https://www.expasy.org/resources/compute-pi-mw; Accessed on 21 Novembe 2021) online platform. Gene family profiles were constructed using the maximum likelihood method (repetitions: 10,000) using Kidio’s online network analysis platform (https://www.expasy.org/resources/compute-pi-mw Accessed on 21 Novembe 2021) for AUREO, respectively, and iTOL (https://www.omicshare.com/tools/Home/Soft/aa2tree, accessed on 21 Novembe 2021) was used to optimize the phylogenetic analysis online (https://itol.embl.de/, accessed on 21 Novembe 2021). The Gene Structure Display Server 2.0 (http://gsds.gao-lab.org/index.php, accessed on 21 Novembe 2021) [58] was used to import the evolutionary tree files into the program while adjusting the Intron rescale to the same length to map three brown algae (*S. japonica*, *E. siliculosus,* and *U. pinnatifida*) (Figure 3).

SOPMA (https://npsa-prabi.ibcp.fr/cgi-bin/npsa_automat.pl?page=npsa_sopma.html, accessed on 21 Novembe 2021) [59] was used to predict the secondary structure information of the AUREO family members, and the minimum free energy of the secondary structure was also used to predict the secondary structure of the AUREO family members. The secondary structure of the AUREO family members was predicted by RNA structure tools (http://rna.urmc.rochester.edu/RNAstructureWeb/Servers/Predict1/Predict1.html, accessed on 21 Novembe 2021), while the secondary structure was predicted by the minimum free energy (MFE) structure. The schematic diagram of the protein structural domain was completed by DOG 2.0 (http://dog.biocuckoo.org/, accessed on 21 Novembe 2021).

Phosphorylation site prediction was performed using NetPhos (https://services.healthtech.dtu.dk/service.php?NetPhos-3.1, accessed on 21 Novembe 2021) [60] software. Transmembrane structure prediction analysis using (https://services.healthtech.dtu.dk/service.php?TMHMM-2.0, accessed on 21 Novembe 2021) software. Subcellular structure prediction localization using WolfPSORT online software (https://wolfpsort.hgc.jp/, accessed on 21 Novembe 2021). By intercepting only 2000 bp upstream of the AUREO start codon in three brown algal genomes (*E. siliculosus*, *S. japonica*, *U. pinnatifida*), we used PlantCARE (http://bioinformatics.psb.ugent.be/webtools/plantcare/html/, accessed on 21 Novembe 2021) to predict its cis-acting elements and then mapped the promoter cis-elements by TBtools after screening (Figure 4). PI and MW values of AUREO were obtained via the website Expasy (https://www.expasy.org/resources/compute-pi-mw, accessed on 21 Novembe 2021).

### 4.3. S. japonica Materials Culture and Expression Analysis of Sj AUREOs

The gametophytes of *S. japonica* were preserved in our laboratory under special conditions, i.e., 4 °C, 24 h continuous LED red light (Dominant wavelength: 639.2 nm) below 5 μmol photons m^−2^s^−1^ and natural sterile seawater enriched with NO3--N of 5 mg/L and PO43--P of 1 mg/L. The gametophytes keep vegetative growth slow by mitosis and form the gametophyte filament clone, and the gametophyte clone of this state is called “dormancy” or delayed gametophyte. To obtain gametophyte clones with sufficient biomass, the dormant gametophyte clones were transferred to conditions suitable for the rapid vegetative growth of gametophytes. The specific culture conditions were as follows: 11–14 °C, continuous LED red light (Dominant wavelength: 639.2 nm) of 80 μmol photons m^−2^s^−1^ and f/2 culture medium. When the biomass in each batch of bottles reached approximately 40–60 mg, the bottles were left to stand, the top layer of clear culture medium was poured off, and the three bottles were mixed into one. They were then placed under the specific conditions mentioned above to make the gametophytes enter a dormant state. To investigate the light-dependent expression pattern of *Sj* AUREO, the dormant gametophytes were transferred into regularly illuminated by white light condition: 80 μmol photons m^−2^s^−1^; 12 h of white light followed by 12 h of darkness. Gametophyte was pre-adapted to three-day photoperiodic changes before conducting experiments. The experiment was confirmed to start at 8:00 am the following day after pre-adaptation. In other words, during the experiment, 8:00–16:00 is the normal light time, and 16:00 to 8:00 the next day is the dark cycle. Expression levels were measured at 4 h intervals, and the experiment continued for 44 h. At each collection, the gametes were filtered through an 800 mesh filter cloth, and residual water was absorbed from the side without gametes using blotting paper. The gametophytes were then scraped from the filter cloth with a sterilized spoon and placed in 1.5 mL sterilized tubes, rapidly snap frozen in liquid nitrogen, and stored at −80 °C.

Samples were taken at four hours intervals for RT-qPCR analysis. Specific qRT-PCR primers (Appendix A) were designed for the *Sj* AUREO genes by NCBI Primer-BLAST (https://www.ncbi.nlm.nih.gov/tools/primer-blast/, accessed on 10 September 2021) [61]. β-actin (GenBank Accession: FJ375360) was selected as the housekeeping gene because it was shown to be stably expressed under light conditions in a previous study [30]. The SPARKeasy. Polysaccharide Polyphenol/Complex Plant RNA Rapid Extraction Kit (SparkJade, Qingdao, China) and SPARKscript II RT kit with gDNA Eraser (SparkJade, Qingdao, China) were used to extract RNA and prepare cDNA. ChamQ SYBR Color qPCR Master Mix (Vazyme, Nanjing, China) and Thermal Cycle Dice™ Real Time System (Takara, Kusatsu, Japan) were used for fluorescence quantification PCR. Expression levels were determined by averaging the expression values of three replicates for each experimental condition.

### 4.4. Expression Pattern of AUREOs during Gametophyte Development

The expression pattern of the AUREO gene in the different developmental stages of *S. japonica* gametophyte was calculated using the public available transcriptome data of *S. japonica*. Raw-reads data were deposited in the NCBI Sequence Read Archive (SRA) with the accession number PRJNA816677 (Paper not published but data uploaded). In the experiment for transcriptome analysis, dormant gametophyte clones were cultured under conditions suitable for their growth and development. The gametophytes were sampled on day 0 (gametophytes were in a dormant state), day 3 (gametophytes were in a fast vegetative growth phase), day 6 (gametophytes underwent vegetative growth that slows and transfers into the developmental state), and day 9 (the partial cells of gametophyte formed the oogonium), where each set has three repetitions. RNA-seq libraries for these samples were prepared, and paired-end sequencing (PE) was performed on the Illumina HiSeq 2000 sequencing platform. The RPKM (reads per kilobase of transcript per million mapped reads) was calculated for each gene using the cleaned data after quality control. Differential expression analysis was performed using the DEGSeq R software (https://bioconductor.org/packages/release/bioc/html/DEGseq.html; Version number: Bioconductor version; Creator: Wang Likun; Beijin, China) package. Trend analysis was performed on all expressed genes using STEM software. Genes with an adjusted *p*-value < 0.05 and a log_2_ (foldchange) > 2 were considered as significantly differentially expressed compared to the control.

## 5. Conclusions

In conclusion, based on the published algal genome data, the AUREO gene family of 7 species of algae was identified with bioinformatics methods in this study. The gene structure of AUREO genes and upstream cis-regulatory elements were analyzed. Then the protein sequences coded by AUREO genes and the physicochemical properties of these proteins were predicted. Built on the phylogenetic tree constructed using AUREO protein sequences, the quantitative changes and subgroup classification of the AUREO gene family in different algae were explored, and the evolution of the AUREO gene family in different algae was discussed. The study further focused on the important economic seaweed, namely *S. japonica*, to analyze the expression of AUREOs in distinct developmental stages of the gametophyte. The result indicated that the expression of *Sj* AUREO varied significantly during the different development stages of *S. japonica* gametophytes, and *Sj* AUREO existed in a light-dependent circadian expression pattern. This may indicate that blue light affects gametophyte development through AUREO as a light signal receptor. The data and information achieved in this study lay a good foundation for future studies of the AUREO gene family as blue light receptors in regulating algal growth, development, and reproduction.

## Data Availability

RNA-Seq data can be found with accession number PRJNA816677. The RNA-Seq data is publicly available at National Center for Biotechnology Information. The other data presented in this study are available in Appendix A.

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
