# Peer review of "Genome-Wide Identification and Analysis of the Aureochrome Gene Family in Saccharina japonica and a Comparative Analysis with Six Other Algae"

_plants, 2022, doi:10.3390/plants11162088_

Round 1

Reviewer 1 Report

General:

The manuscript by Wu et al. performed a comparative analysis of 44 aureochrome genes in 7 algae, and discussed the evolution of aureochromes. Authors, then, focused on 9 aureochromes in S. japonica (SjAUREOs), and extracted their expression levels from transcriptome in different developmental stages. They also investigated expression map of SjAUREOs under periodic light. 

The comparative analyses of aureochrome genes were performed by the sequences corrected from databases, and detailed data such as genomic location of each gene etc. are not intrinsic in this study, and preferably indicated in the supplementary information. Similarly, secondary structures, phosphorylation sits, subcellular localization, and cis-acting elements predicted by online platform are preferred to be shown in the supplemental information. Similar figures (Fig. 1B, Scheme 1, and supplementally Fig. 1) should be reorganized and summarized with detailed explanation in the legend. 

Reviewer couldn’t evaluate Scheme 1, Fig. 5 and Fig. 6, because the characters are too small to read even at maximum magnification. These figures should be re-organized.

From the results of study, authors concluded the SjAUREOs play the important role as a blue light receptor in regulating the growth and development of S. japonica (L534-L536). However, this conclusion can’t be led from the analyses of sequences comparison, expression levels, and expression map.

Specific:
L127: Species name should be italic

Like other AUREO (L544-549).

V. frigida AUREO should be written as VfAUREO.

E. siliculosus AUREO should be written as EsAUREO.

Fig. 1 legend: There are only two panels (A and B) in Fig. 1; colour

L221: tegment

L239: four4, five5

L245: ofwith

L290: abbreviation FPKM should be explained

L442: duplicate “Structural and phylogenetic analysis of AUREO”

Author Response

Replies to Reviewer #1:

Point 1:

Reviewer: Detailed data such as genomic location of each gene etc. are not intrinsic in this study, and preferably indicated in the supplementary information. Similarly, secondary structures, phosphorylation sits, subcellular localization, and cis-acting elements predicted by online platform are preferred to be shown in the supplemental information.

Reply: We agree with the reviewer's opinion. We revised the original Table.1 and Table.2 to Supplementary Table 1 and Supplementary Table 2. Re-modified the titles of other supplementary tables.

Modification in the manuscript:

(L153,173) Move Table .1 and Table .2 out of the main text and place them in Supplementary Materials.

(L136,152,163,449,451,514) Revised references to supplementary tables in the article: Table.1- Suppl. Table 1, Table.2- Suppl. Table 2, Suppl. Table 1/2/3/4/5- Suppl. Table 3/4/5/6/7

Point 2:

Reviewer: Similar figures (Fig. 1B, Scheme 1, and supplementally Fig. 1) should be reorganized and summarized with detailed explanation in the legend.

Reply: In this part, our description is not clear enough and there are some logo errors. We have added specific explanations to the pictures and corrected the logo errors.

Modification in the manuscript:

(L181-194) Fig.1

Figure 1. (A) Schematic diagram of bZIP and LOV domains in S. japonica AUREOs. (B) Comparison of the structural domain characteristics of Sj AUREO and Arabidopsis thaliana bZIP (At bZIP). According to the prediction results of Pfam on the positions of the two domains of each AUREO gene, we have drawn the structural diagrams of nine Sj AUREOs. The length of the grey box represents the amino acid length of each Sj AUREO; red represents the bZIP domain; light blue represents the LOV domain. The exact position of each domain is given numerically in the figure. The thirty-four amino acids bZIP feature sequence fragment was intercepted in Fig.1B. Comparing SjAUREOs with G-GBF, C-BZ02H and S-GBF in AtbZIP (where G/C/S represents the bZIP grouping in the Arabidopsis study, followed by the name of the gene). The seven amino acids in the bZIP domain that can bind to DNA are marked with black boxes. The ZIP domain consists of heptad repeats of leucine (L) or related hydrophobic aa [44]. Located at the 19th, 26th and 33rd amino acids, respectively. This is simulation uses the default colour for alignments in ClustalX. Different categories of amino acids are labelled separately with different types of colours. Each residue in the alignment is assigned a colour if the amino acid profile of the alignment at that position meets some minimum criteria specific for the residue type. Conservation: conservation of total alignment less than 25% gaps; Quality: alignment quality based on blosum62 scores; Consensus: Characteristics of each repeat of AUREO; Occupancy: number of aligned positions.

(L208-213) Scheme 1

Scheme 1 and supplementally Fig. 1 is the same figure, which we normalize to Scheme 1

Scheme 1. Characteristics of each repeat of AUREO. Comparison of the bZIP (Left) and LOV (Right) domains of AUREO proteins from S. japonica and other six algae. i.b.Fig.1B. Blue: Hydrophobicamino acid; Red: Positive charge amino acid; Magenta: Negative charge amino acid; Green: Polar amino acid; Yellow: Prolines amino acid; Orange: Glycines amino acid; White: Unconserved

Point 3:

Reviewer: Reviewer couldn’t evaluate Scheme 1, Fig. 5 and Fig. 6, because the characters are too small to read even at maximum magnification. These figures should be re-organized.

Reply: Here we provide clear versions of Scheme 1, Fig.5 and Fig.6. The original images are also packaged in a .zip file

Modification in the manuscript: For the clarity of Fig.5 and Fig.6, reviewer 2 also raised the same question, please see the fifth and second questions in Replies to Reviewer #2 for a clear picture, and we have also re-provided clear pictures in the article information

Point 3:

Reviewer: From the results of study, authors concluded the Sj AUREOs play the important role as a blue light receptor in regulating the growth and development of S. japonica (L534-L536). However, this conclusion can’t be led from the analyses of sequences comparison, expression levels, and expression map.

Reply: We agree with the comments made by reviewer 1 on this part of the content. This part of the study is unable to draw firm conclusions to prove that it is related to gametophyte development. We revise the conclusions that cannot be deduced from the results and the too absolute discussion part of the article. Based on the expression pattern of AUREO obtained from qPCR and RNA-seq data, we speculated that the blue light receptor AUREO may have a certain relationship with gametophyte development. We retain this view because the current research on AUREO is not sufficient, and the research on SjAUREO is almost a blank, and our analysis can provide a preliminary understanding. We believe that it is necessary to reserve this conclusion to guide future research

Modification in the manuscript:

(L559-566) The result indicated that the expression of Sj AUREO varied significantly during the different development stage of S. japonica gametophytes and Sj AUREO existed a light-dependent circadian expression pattern. This may indicate that blue light affects gametophyte development through AUREO as a light signal receptor. The data and information achieved in this study lay a good foundation for future studies of the AUREO gene family as blue light receptors in regulating algal growth, development and reproduction.

Detailed pictures we placed in the accompanying Word document uploaded. The revised article and images were sent to the editorial office in a revised format

Reviewer 2 Report

The manuscript by Wu et al. identified and characterized Aureochromo genes in S. japonica and make comparisons with other six algae.  The paper need a minor revision at current stage. 

1. How the classification of SjAUREO1a/1b/1c/1d/1e,  SjAUREO2, SjAUREO3, etc. been determined? 

2. Add error bar to Figure 6.

3. In section 4.4, please clarify replicates number for RNAseq data and criterion used for differential expression genes determination. 

4. Gene names should be italicized and protein names should be non-italicized. Also check the species Latin names, some of them are not in italic. 

5. Font of Figure 5 and Figure 6 need to be adjusted to be more reading friendly. The labels are not visible now. I suggest to use different colors for each of the SjAUREO gene in Figure 6.

Author Response

Replies to Reviewer #2:

Point 1:

Reviewer: How the classification of Sj AUREO1a/1b/1c/1d/1e, Sj AUREO2, Sj AUREO3, etc. been determined?

Reply: The classification of Sj AUREO is mainly based on the classification and naming of AUREO in two model algae, Phaeodactylum tricornutum and Ectocarpus siliculosus. In our study, we fully followed the earlier conclusions in these two model algae. Because E. siliculosus and Saccharina japonica are closely related and both belong to Phaeophyceae, we firstly used the study of E. siliculosus as a blueprint to classify and name the five genes in S. japonica (Sj AUREO1a/2 /3/4/5). Finally, we classify and name the remaining five Sj AUREO (1b/1c/1d/1e) according to the phylogenetic tree. We believe that such a classification is helpful for recognizing and distinguishing different AUREO genes in Saccharina japonica. There is currently no systematic classification and nomenclature for the AUREO family. Therefore, we prefer to classify the AUREOs in S. japonica according to the AUREOs that have been classified and named in previous studies.

Modification in the manuscript: no modification

Point 2:

Reviewer: Add error bar to Figure 6.

Reply: We have revised the picture that was not clear enough. Here we also provide a scalable version of Fig 6. We have added error bars and colored the different AUREO genes. We also think that such changes will help future readers can better understand the gene expression pattern of AUREO.

Modification in the manuscript: (L308) Removed the original unclear image and re-provided a scalable version.

Fig. 6

Point 3:

Reviewer: In section 4.4, please clarify replicates number for RNAseq data and criterion used for differential expression genes determination.

Reply: We supplemented the article with specific parameters for RNAseq analysis

Modification in the manuscript:

New addition

(L534) “where each set has three repetitions”

(L539-541) “Trend analysis was performed on all expressed genes using STEM software. Genes with an adjusted p-value < 0.05 and a log2(foldchange) >2 were considered as significantly differentially expressed compared to the control.”

Point 4:

Reviewer: Gene names should be italicized and protein names should be non-italicized. Also check the species Latin names, some of them are not in italic.

Reply: We rechecked all species Latin names and protein names in the article, and corrected errors to correct italics. Details are in the "Specific" section

Modification in the manuscript:

In the reworked text marked with the revised format

Modify similar parts: “SjAUREO” - ”Sj AUREO” “S. japonica” - ” S. japonica

Point 5:

Reviewer: Font of Figure 5 and Figure 6 need to be adjusted to be more reading friendly. The labels are not visible now. I suggest to use different colors for each of the Sj AUREO gene in Figure.

Reply: We have re-provided clear Fig. 5 and Fig. 6 as an attachment.

Modification in the manuscript:

Fig. 5

Detailed pictures we placed in the accompanying Word document uploaded. The revised article and images were sent to the editorial office in a revised format.

Round 2

Reviewer 1 Report

Comments:

I learned that the authors constructively revised the manuscript. However, the characters in scheme 1, figure 5 and 6 are still small to read in the text, which was also pointed out by the reviewer #2. Can the authors access the original data?  As an example, I simply attempt to revise panels as shown in the word file.

Is the heading OK?: “4.2. AUREO Structural and Phylogenetic Analysis Structural and phylogenetic Analysis of AUREO” (L468)

Author Response

We reworked and uploaded three unclear images in PDF format (Figure 5 and 6, scheme 1 are placed in the PDF file in order). At the same time, the title of 4.2 in the article was re-modified.

Modification in the manuscript: (L468) “AUREO Structural and Phylogenetic Analysis Structural and phylogenetic Analysis of AUREO” - “Structural and Phylogenetic Analysis of AUREO
